# Analysis of Tonsil Tissues from Patients Diagnosed with Chronic Tonsillitis—Microbiological Profile, Biofilm-Forming Capacity and Histology

**DOI:** 10.3390/antibiotics11121747

**Published:** 2022-12-03

**Authors:** Marina Kostić, Marija Ivanov, Snežana Sanković Babić, Zvezdana Tepavčević, Oliver Radanović, Marina Soković, Ana Ćirić

**Affiliations:** 1Institute for Biological Research “Siniša Stanković”—National Institute of Republic of Serbia, University of Belgrade, Bulevar despota Stefana 142, 11060 Belgrade, Serbia; 2Clinic for Otorhinolaryngology, Clinical Hospital Centre Zvezdara, Preševska 31, 11000 Belgrade, Serbia; 3Department of Pathology, School of Dental Medicine, University of Belgrade, 11000 Belgrade, Serbia; 4Institute for Veterinary Medicine of Serbia, Janisa Janulisa 14, 11000 Belgrade, Serbia

**Keywords:** *Staphylococcus aureus*, biofilm, histology, chronic tonsillitis

## Abstract

Chronic tonsillitis (CT) is a global health issue which can impair patient’s quality of life and has an important socioeconomic impact due to the nonrational use of antibiotics, increased antimicrobial resistance and frequent need for surgical treatment. In order to isolate and identify the causing agents of CT, a total of 79 postoperative palatine and adenoid tissue samples were obtained from the ENT Clinic, KBC Zvezdara, Belgrade, Serbia. Culture identification was performed by MALDI-TOF MS and the *Staphylococcus aureus* isolates were tested for biofilm forming capability and antibiotic susceptibility. Additionally, a histological examination of palatine and adenoid tissue was performed in order to detect the presence of CT-causing bacteria. The slight majority of participants were females with median age of 28 years for adult patients (group I) and 6 years for children (group II). Analysis of the incidence of bacteria isolated from tissue samples in both groups showed the highest prevalence of *S. aureus*, *Streptococcus oralis* and *Streptococcus parasanquinis*. In addition to interfollicular hyperplasia, colonies of species *S. aureus* were detected in histological material. The presence of biofilm might be the reason for the recurrence of infection. Therefore, searching for a new treatment of CT is of great importance.

## 1. Introduction

Upper respiratory tract (URT) infections include inflammation of the nose (rhinitis), sinuses (sinusitis), middle ear (otitis media), pharynx (pharyngitis), tonsils (tonsillitis) and larynx (laryngitis). The microbiota of different regions of URT consists of a large number of diverse microorganisms. The most commonly isolated bacteria belong to the genera *Streptococcus*, *Neisseria*, *Haemophilus*, *Moraxella*, *Staphylococcus*, *Corynebacterium*, *Propionibacterium*, *Prevotella*, and *Porphyromonas* [1,2,3]. Among them, the most common causes of infections are: *Streptococcus pneumoniae*, *S. pyogenes*, *Haemophilus influenzae*, *Moraxella catarrhalis*, and *Staphylococcus aureus* [3,4,5,6,7]. The imbalance in the URT microbiota is associated with the increase in the number of invasive pathogenic bacteria, leading to inflammation and infection. Infection and development of the disease occurs after colonization of the mucous membrane of the URT by pathogenic bacteria [2]. Colonization by different bacteria is a continuous process on the mucous membranes of the nose and throat in both children and adults.

Waldeyer’s lymphatic ring is a barrier to the penetration of pathogens into the respiratory and digestive tract. The Waldeyer’s ring includes: the pharyngeal tonsil (*tonsilla pharyngea*) or adenoid, which is attached to the upper wall of the nasal part of the pharynx; tubal tonsils (*tonsilla tubaria*) located in the immediate vicinity of the pharyngeal openings of the Eustachian tubes; palatine tonsils (*tonsilla palatina*) located in the oral part of the pharynx, oropharynx, on the side walls between the anterior and posterior palatal arches and the lingual tonsil (*tonsilla linqualis*) located on the last third of the tongue [8]. It has a role in humoral (synthesis of a large number of immunoglobulins) and cellular immunity (activation of T and B lymphocytes) [9]. The palatine tonsils play a role in initiating an immune response against antigens that enter the body through the oral cavity, while the adenoid protects the nasopharyngeal mucosa from airborne pathogens. These structures have the greatest immune activity in children aged 3–10 years. The palatine tonsils are known to produce five isotypes of immunoglobulins (IgA, IgM, IgE, IgD, and IgG), of which IgA is considered the most important because secretory IgA (SIgA) is a key component of the URT mucosal immune system [10]. The tonsils and URT are protected from microbiological, allergological and other agents as long as the tonsils’ epithelium is intact. When the epithelium is disrupted, the function of the tonsils is disturbed, and lesions appear which are often filled with purulent contents. In diseased tonsils, the infection can spread lymphatically to regional lymph nodes. Chronic tonsillitis (CT) is a long-term infection that occurs as a result of multiple repeated infections of the tonsils. The pathogenesis of the disease is a consequence of several repeated episodes of acute tonsillitis or as a result of a persistent infection that leads to chronic inflammation that is long-lasting and slowly progressing. Inflammation can affect all lymphatic structures belonging to the Waldeyer’s ring. The palatine tonsils and adenoids are most often affected by inflammation [11]. There are several ways to categorize this infection: (a) CT can be of bacterial, viral and fungal etiology, depending on the infectious agent underlying the chronic inflammation; (b) according to macroscopic and microscopic characteristics, CT can be hypertrophic or atrophic; and (c) in relation to age, CT can be divided into CT in children and CT in adults. Tonsillitis is the third most common among ENT (ear, nose, and throat) infections in the general population, after rhinopharyngitis and otitis. In France, about nine million cases of tonsillitis are diagnosed each year, while in Spain that number is four million [12]. The prevalence of CT is up to 11.7% in United States, while the number of tonsillectomies annually increases. Recurrent throat infections or sleep-disordered breathing are indications for more than 530,000 tonsillectomies which are performed in children and adolescents in the US every year. Children with recurrent tonsillitis have significantly worse health status and physical functioning compared to healthy children of the same age [13,14].

Biofilm is considered the main microbiological factor that leads to chronic infections because it protects bacteria from the host’s defensive response and leads to the emergence of resistance to the applied therapy [15]. The tonsillar tissue and adenoids are predisposed to biofilm formation due to cryptic tissue structure and direct, repeated exposure to respiratory bacterial pathogens [16]. *S. aureus* in the form of biofilm can lead to the occurrence of chronic infections in the URT, including CT [17,18]. The nasal and oral cavity may act as the starting points for the spread of *S. aureus* infection to other sites in the body and for the development of systemic infections [19]. Except *S. aureus*, several other respiratory pathogens, such as *Haemophilus influenzae*, *Klebsiella pneumoniae* and *Streptococcus pneumoniae*, can persists predominantly intracellularly in the form of biofilm in adenoids and palatine tonsils [2,16,20]. 

Given the importance of the biofilm in infection establishment, the emergence of resistant strains and difficulties in eradicating them, the aim of this study was to identify bacteria that cause CT, determine their sensitivity to commercial antibiotics, to examine their ability to form biofilm as well as to histologically confirm infection and the presence of bacteria in the tissue itself.

## 2. Results

### 2.1. Demographic Data of Operated Patients with Chronic Tonsillitis

A total of 75 CT patients participated in the study, of which 40 were female (53.33%) and 35 (46.67%) were male patients (Table 1). Patients were divided into two groups based on age. The first group consisted of CT patients older than 16 years (Group I), while the second group consisted of CT patients younger than 16 years (Group II). Group II patients are divided into special groups based on the type of tonsillectomy. Namely, group IIa includes patients with only palatine tonsils surgically removed; group IIb includes patients who underwent adenoidectomy in addition to the palatine tonsils removal, and finally, group IIc includes patients in which only adenoidectomy was performed. Group I of operated patients mainly consisted of females (60%), with an average age of 28.13 years (Table 1). Group II of operated patients consisted of 23 males (51.11%) and 22 females (48.89%), average age 5.56 years. Group IIa consisted of 11 males and 16 females; 4 patients of group IIb were male; while group IIc consisted of 8 male and 6 female patients. The average age by groups was as follows: group IIa—6.12 years; group IIb—5 years and group IIc—4.6 years (Table 1).

### 2.2. Clinical Characteristics of Operated Patients with Chronic Tonsillitis

Based on the anamnestic data, the most common symptom in all patients (group I) with moderately severe (56.66%) and severe problems (43.33%) was a sore throat. Other symptoms manifested in most cases (>50%) with moderate to severe symptoms were difficulty swallowing (56.66%), the presence of thick secretions in the throat (50%), and bad breath (66.66%). All patients took medications from the group of beta lactam antibiotics and cephalosporin’s of the third generation as part of drug therapy for the treatment of inflammation. Analysis of oropharyngoscopic status showed that 13 patients required surgery due to hyperemic, hypertrophic, cryptic tonsils, while another 17 patients had asymmetric, uneven tonsils with palatine arch hyperemia as an indication for surgery. Based on the data collected by the questionnaire, the prevalence of symptoms and their severity differ within group II. The most common symptom in group IIa was shortness of breath, as many as 15 patients (55.55%) stated that they had unbearable breathing difficulties. Likewise, the presence of a thick secretion in the throat was indicated as a problem for less than half of the operated patients (40.74%). In group IIb, the only symptom with high incidence (75%) was snoring. Shortness of breath, the presence of thick secretions in the nose and snoring are the symptoms that were most common in patients with group IIc. The analysis of the group I oropharyngoscopic status showed that 13 patients required surgery due to hyperemic, hypertrophic, cryptic tonsils, while another 17 patients, as an indication for surgery, in addition to the above-mentioned symptoms, also had asymmetric, uneven tonsils with hyperemia of the palatal arches. Analysis of oropharyngoscopic status showed that 25 patients (92.59%) of group IIa and 4 (100%) patients of group IIb required surgery due to hyperemic, hypertrophic, cryptic tonsils. On the other hand, adenoidectomy was performed in all patients in group IIb, because asymmetric lobed tonsils with hyperemia of the palatine arches, enlarged adenoid, as well as blurred and non-transparent eardrums were observed as the indications for surgery. In group IIc, oropharyngoscopic status was positive in all patients, while otoscopic status was positive in 11 of 14 patients (Table 2).

### 2.3. Identification of Bacterial Pathogens

A total of 82 bacteria were isolated and identified from all samples, 35 isolates from Group I tissues and 47 from sampled Group II tissues (Table 3, Appendix A). A total of 8 aerobic and 8 facultative anaerobic species from the genera *Streptococcus*, *Staphylococcus*, *Micrococcus*, *Rothia*, *Enterobacter*, and *Stenotrophomonas* were identified. Analysis of the bacteria isolated from palatine tonsil tissue samples in Group I patients showed the highest prevalence of *Staphylococcus aureus* (29%), followed by other G (+) bacteria *Streptococcus oralis* (20%) and *S. parasanquinis* (18%), while only one G (-) bacterium was isolated, *Enterobacter cloacae* (6%) (Figure 1). In Group II patients the analysis of bacteria isolated from the tissues of the palatine tonsils and adenoids showed the highest prevalence of *S. oralis* (43%), followed by *S. parasanquinis* (22%) and *S. aureus* (15%) (Figure 2A–D). The only G (-) bacterium isolated from adenoids was *Stenotrophomonas maltophilia* (Figure 2D). In both groups, species belonging to group A β-hemolytic streptococci were the most frequently isolated ones. In several samples of palatine tonsil tissue, *S. aureus* was co-isolated with *S. oralis* (4 samples), and with *S. dysgalactiae* in one sample. Additionally, in one sample of palatine tonsil tissue, in addition to *S. aureus*, two other species were co-isolated, namely *S. oralis* and *Rothia mucilaginosa*. Therefore, there were six tissue samples with more than one type of bacteria isolated. 

#### 2.3.1. Biofilm Formation

Among three the most isolated bacteria species (*S. aureus*, *S. oralis* and *S. parasanquinis*), only *S. aureus* isolates demonstrated the ability to form biofilm, so the authors selected only these isolates for further testing. All isolates of *Staphylococcus aureus* were tested for their biofilm formation ability. A biofilm forming ability was moderate for 2 (11.8%) and strong for 15 (88.2%) *S. aureus* isolates. None of the isolates showed weak ability to form a biofilm (Table 4, Appendix A).

Interpretation of biofilm production: OD ≤ ODc—none; ODc ˂ OD ≤ 2 × ODc—weak; 2 × ODc ˂ OD ≤ 4 × ODc—moderate; 4 × ODc ˂ OD—strong.

#### 2.3.2. Susceptibility Testing Data

The susceptibility of *S. aureus* clinical isolates to different groups of antibiotics (penicillin, cephalosporin, macrolides, aminoglycosides, and polyketides) was examined. Clinical isolates showed the highest (100.0%) resistance to each of penicillin, ampicillin and amoxicillin, while the lowest (11.8%) antibiotic resistance was recorded for macrolides class of antibiotics and tetracycline (Table 5). Towards all other tested antibiotics *S. aureus* demonstrated high susceptibility (Table 5). 

As we mentioned in Section 2.2, all patients took medications from the group of beta lactam antibiotics and cephalosporin’s as part of drug therapy. Based on questionnaire the most prescribed drugs for the CT treatment were amoxicillin with clavulanic acid (beta lactam antibiotic) and cefixime (cephalosporin). Therefore, these two antibiotics were chosen for further susceptibility testing (Table 6). *S. aureus* isolates proved to be more sensitive to amoxicillin with clavulanic acid with MIC values in the range of 0.001–0.013 mg/mL (Table 6). The MIC/MBC values for cefixime were slightly higher (0.003–0.014 mg/mL/0.006–0.028 mg/mL, respectively). 

### 2.4. Analysis of Palatine and Adenoid Tissue

Tissues of the palatine tonsils were obtained from the surgically resected material for histological analysis in order to detect the presence of CT-causing bacteria. The presence of enlarged follicles with brown bodies (intrafollicular hyperplasia) was observed on the obtained sections of tonsil tissue (Figure 3A). Only a part of the epithelium can be clearly seen on the histological section because the epithelium is disrupted due to the inflammatory process (Figure 3A). In Figure 3, the cross section 3B and 3C at different magnifications (40× and 100×, respectively) the colony of the G (+) bacteria *S. aureus* was observed, which confirms that this species is one of the causes of CT. The presence of microbial biofilm on the histological sections of the tonsillar tissue was not detected. The study analyzed and presented only histological sections of the tissues of diseased patients without the possibility of comparison with healthy tissue because such tissue is not subject to tonsillectomy.

## 3. Discussion

This is the first study conducted in Serbia that examined the microbiological profile of the tonsillar tissue core. Chronic tonsillitis has a negative impact on quality of life. Patients most often complain on the frequent inflammation of the throat and purulent angina, recurrent peritonsillar abscesses, tingling, burning, and a constant feeling of a foreign body in the throat, bad breath, difficulty breathing and/or swallowing, persistent chronic infection of the nose, paranasal cavities, etc. The symptoms are especially pronounced in children, where often lead to chronic snoring or even suffocation during sleep. These symptoms are accompanied by poor nutrition and development of the child. Nasopharyngeal obstruction disrupts the ventilation of the eardrum and can lead to the development of secretory otitis with consequent deafness [21]. Based on data conducted in our study (Table 2), the most common symptom was sore throat (Table 2). Many authors found that the most common symptoms for surgical removal of tonsillar tissue were: airway obstruction [22,23], sleep apnea [24] and in other cases sore throat, fever, pharyngitis, stomatitis, and tonsillar tumor [25]. Haidara et al., in a one-year follow-up of 315 patients, showed that the most common symptom of CT was swallowing pain—odynophagia (88, 63%), while other incidence-based symptoms were as follows: fever (86, 27%), snoring (38, 43%), ear pain (37, 65%), excessive salivation (20.39%), difficulty swallowing (15, 25%), sleep apnea syndrome (14, 12%), and shortness of breath (10, 20%) [12]. 

According to Hibbert et al. [26], in 55 (94%) of the cases with acute tonsillitis children were prescribed an antibiotic, which is in accordance with our data obtained by filling out the questionnaire (Table 2). In support of the fact that in recent years there has been a higher number of resistant strains due to excessive use of antibiotics, Cavalcanti et al. [15] reported that 83.6% of *Staphylococcus aureus* isolates from tonsil tissue was resistant to penicillin and 13.6% to amoxicillin with clavulanic acid. Additionally, the present study demonstrated that *S. aureus* isolates were resistant to penicillin (Table 5). The ineffectiveness of penicillin has led to increased use of other antibiotics such as β-lactam inhibitors and cephalosporins [15,27]. Antibiotic resistance can be explained by the impossibility of antibiotic penetration and action in the tonsil core, (especially if the bacterial cells are covered with an biofilm extracellular matrix), the resistance of strains to typical antibiotic treatments due to the constant use of antibiotics due to recurrent infections and the prevalence of biofilm formation [28]. Additionally, one of key factors to antibiotic resistance against penicillin group of antibiotics is beta-lactamase production. These enzymes are produced by Gram (+) bacteria extracellularly, and by Gram (-) bacteria in the periplasmic space. Β-lactamases can inactivate almost all β-lactam antibiotics by binding covalently to their carbonyl moiety and hydrolyzing the β-lactam ring [29]. 

Oropharyngoscopic status showed that bilateral hypertrophy of the palatine tonsils was present in 82.35% of operated cases [12], which is consistent with the present study (Table 2). The data suggest that a number of pathogenic microorganisms isolated from infected tonsils may influence the development of tonsillar hypertrophy [30]. Torretta et al. [17], showed that there is a significant association between bacteria isolated from tonsil tissue that can form biofilm and tonsil hypertrophy. 

Recurrent infection results in CT, where tonsillar hypertrophy leads to respiratory obstruction [5]. In the present study, the most frequent bacteria were *S. aureus* and *S. oralis* (Table 3). Our results showed prevalence of *S. oralis* in children, which can be explained by the outbreaks of tonsillitis which is common in children of all ages, especially in the school-aged children in which is crowding common [31]. Additionally, *S. oralis* is known to be the first to colonize enamel, often interacting with periodontal pathogens such as *Porphyromonas gingivalis* which is the causative agent of chronic periodontitis [32]. Repeated and untreated Streptoccocal infection can lead to the colonization of some other pathogenic bacteria such as *S. aureus* in adults (Table 3). Additionally, in immunocompromised patients, it can cause bacterial endocarditis, respiratory diseases, and can also cause streptococcal septicemia [33]. Jeong et al. [34] identified *S. aureus* (30.3%), *Haemophilus influenzae* (15.5%), and *S. pyogenes* (14.4%) as the most commonly isolated bacteria in patients with CT. Microbiological analysis of palatine tonsil and adenoid tissues, showed that species of the genera *Bacterioides* and *Streptococcus* are present in the crypts of the tonsils, *H. influenzae* was observed in the tissue itself, while species of the genera *Fusobacteria*, *Pseudomonas*, and *Burkholderia* were isolated from surface tissue layers. These results suggest that hyperplasia of the adenoids and palatine tonsils is probably the result of the presence of several species of pathogenic bacteria, which changes the previous perception that adenotonsillitis is a consequence of infection caused by a single bacterial species that colonizes only the tissue surface [35]. In accordance with the obtained results (Figure 1 and Figure 2), a number of other studies in which the tissue of the palatine tonsils was analyzed, it was shown that the most common bacterial species in the tissues is *S. aureus* [18,27,36]. Galli et al. [36] has shown that all isolated strains from palatine tonsil tissue have the ability to form a biofilm, whereas in the case of adenoid tissue 60% of species had the ability to form a biofilm [36], while in the present study majority of *S. aureus* isolates have strong biofilm-producing capacity (Table 4). Other authors showed that except of *S. aureus*-positive tissue samples, the most common isolates were: *Streptococcus anginosus*, *Streptococcus constellatus*, *S. viridans*, *H. influenzae*, and *H. parainfluenzae* [14,32]. Several studies have identified *S. aureus* as the main cause of tonsillitis, or as a likely co-pathogen with a prevalence of 83% [17,18,37]. 

The presence of *S. aureus* in the tonsillar tissue even after the inflammatory process may be related to its ability to form a biofilm (Figure 3). The presence of biofilm explains the inactivity of antibiotics and thus the recurrence of infection [15]. Chole and Faddis [38] confirmed, by histological analysis, the presence of bacteria in the form of biofilm in the crypts of palatine tonsil tissue. Dense accumulations of G (+) bacteria were observed in the crypts, while inflammatory cells were seen on the periphery of these bacterial colonies. Bacterial colonies were detected by electron microscopy as an amorphous mass immersed in the extracellular matrix, which proved the authors’ presence of biofilm. Bacteria in the biofilm, within the tonsil tissue, are protected from the host’s defensive response and from the action of antibiotics and can synthesize various endotoxins without obstacles. Endotoxins in the crypts of the tonsils lead to chronic inflammation. Additionally, when local environmental conditions are favorable, bacteria in the biofilm become motile, resulting in re-infection [39]. Immunohistological staining of the palatine tonsil section from which *S. aureus* was isolated clearly showed an association between *S. aureus*-specific staining and histological signs of tonsillitis (loose epithelium, tissue necrosis, presence of exudates) [18]. 

## 4. Materials and Methods

### 4.1. Research Methodology

The study was conducted among patients scheduled for tonsillectomy from January to December 2017. The first part of the study included the collection of demographic-epidemiological and clinical data on patients. These data were obtained using pre-structured questionnaire which patients filled in before the surgery. The categorization of the symptoms was performed according to SNOT 20 questionnaire [40], with some modification regarding to disease specificity. The second part included postoperative collection of tonsils and their microbiological and histological analysis. The study included 75 patients diagnosed with CT, who have been admitted to the Otolaryngology clinic of Clinical center Zvezdara for tonsillectomy. This part of the study was approved by the Ethics Committee of the Clinical Hospital Center (KBC) Zvezdara. 

#### 4.1.1. Study Subjects

Between January to December 2017, we recruited outpatients aged 3–40 years who were referred to Otolaryngology Clinic for tonsillectomy because of the CT diagnosis defined as a severe recurrent tonsillar infection (at least three episodes per year) with chronic tonsillar hypertrophy. 

#### 4.1.2. Inclusion and Exclusion Criteria

The inclusion criteria were: long term use of antibiotics in the treatment of CT; throat pain; difficult swallowing and breathing; bad breath; snoring; secretion in the throat and nose; hyperemic, hypertrophic, cryptic tonsils with detritus in the crypts, asymmetric, uneven tonsils with hyperemia of the palatine arches, blurred and non-transparent eardrum. The exclusion criteria were: previous adenoidectomy; acute febrile illness; acute URT infection or antibiotic therapy in the previous 14 days; febrile conditions, concomitant systemic diseases; and craniofacial, neuromuscular, immunological or syndromic abnormalities.

### 4.2. Tissue Sample Collection, Processing and Bacterial Isolation

Tonsillar tissue (adenoids and palatine tonsils) of patients, based on the recommendations of the clinical guide [13] was collected after surgery. After tonsillectomy/adenoidectomy, the tissues samples were transported under sterile conditions to the laboratory up to 20 min from being obtained and processed immediately. Under sterile conditions, the tissue was surgically resected, and the sections were inoculated on nutrient media (blood and chocolate agar). Petri dishes with tissue were incubated at 37 °C and colony development was observed after 24 and 48 h. Colonies were reinoculated on blood agar (Torlak, Belgrade, Serbia) and tryptic soy agar (TSA) to obtain pure cultures at 37 °C for 24 h and these cultures were used for the identification to species level. 

### 4.3. MALDI-TOF Analysis

Identification of isolated strains was performed using the Vitek MS apparatus (bioMérieux, Marcy l’Etoile, France) according to Dubois et al. [41]. The principle of operation of MALDI mass spectrometry includes 3 steps: sample preparation, desorption of the sample and matrix using a laser and protonation/ionization of the sample. The peaks of the obtained profile for a particular isolate were compared with the known protein profiles available in the VITEK database on the basis of which the precise identification of the microorganism was performed. For VITEK MS, reliable identification is achieved if the similarity in the profiles is >90%. 

### 4.4. Biofilm Formation 

*S. aureus* cells were incubated with Tryptic soy broth (TSB) medium (Torlak, Belgrade, Serbia) in 96-well microtiter plates with an adhesive bottom (Spektar, Čačak, Serbia) at 37 °C for 24 h. After incubation and fixation of the cells, plates were dried and stained using 0.1% crystal violet dye (Bio-Merieux, France). Wells were washed again, and air dried, after which 96% ethanol (Zorka pharm, Šabac, Serbia) was added and the OD of stained adherent biofilm was read at 620 nm using a plate reader. The experiment was performed in triplicate. The interpretation of biofilm production was performed according to Stepanović et al. [42]:

OD ≤ ODc—none; ODc ˂ OD ≤ 2 × ODc—weak; 2 × ODc ˂ OD ≤ 4 × ODc—moderate; 4 × ODc ˂ OD—strong.

Optical density cut-off value (ODc) = average OD of negative control + 3 × standard deviation (SD) of negative control.

### 4.5. Antibiotic Susceptibility Testing

Antibiotic susceptibility test of biofilm producing bacteria was performed by using the disc diffusion technique according to EUCAST guidelines [43] and microdillution method [44]. For microdilution method, the results were presented as minimum inhibitory/bactericidal concentrations (MICs/MBCs). Amoxicillin with clavulanic acid (Hemofarm, Vršac, Serbia) and cefixime (Alkaloid, Skoplje, Macedonia) were used as positive controls.

### 4.6. Histological Analysis

Cuts 5 to 10 mm thick were taken from the surgically resected material for histological analysis. Samples were fixed in 10% buffered formalin for 12 h, dehydrated in alcohols of increasing concentration, clarified in chloroform, impregnated and molded with paraffin and cut on a sliding microtome (Leica CM1850, Germany) into 4 µm thick cross-sections, then deparaffinized and stained hematoxilin-eosin (H&E) method [45]. Histological specimens were observed under a microscope (Olympus BX50) to detect, analyze, and describe tissue changes due to the inflammatory process. 

### 4.7. Statistical Analysis

The obtained data were statistically analyzed by using excel. Data were expressed as mean ± SD for qualitative variables, while as a number and percentage for categorical variables.

## 5. Conclusions

Chronic tonsillitis is currently one of the many global public health issues which can severely impair an individual’s quality of life, especially in children. Among others, biofilms are identified as one of the major causes of repeated tonsillitis in both pediatric and adult population. In this work *S. aureus* and *S. oralis* were identified as leading causing agents of chronic tonsillitis. Additionally, we confirmed the *S. aureus* presence in the tonsillar tissue and ability of these isolates to form biofilm. Therefore, in order to reduce tonsillectomy, new therapy strategies need to be developed that will include treatment of biofilms.

## Figures and Tables

**Figure 1 antibiotics-11-01747-f001:**
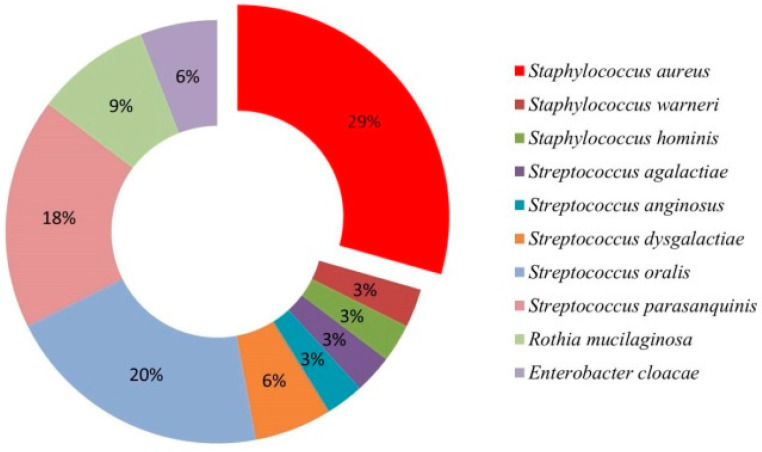
Percentage representation of species in postoperative samples of palatine tonsil tissue of Group I patients.

**Figure 2 antibiotics-11-01747-f002:**
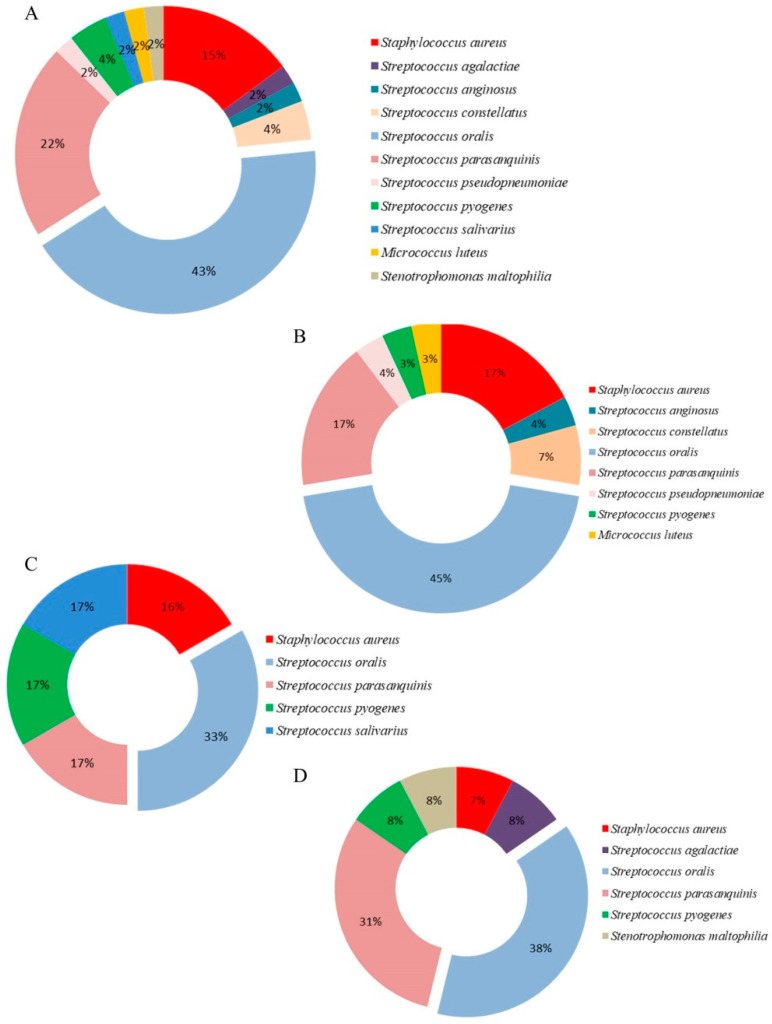
Percentage representation of species in postoperative samples of tonsil tissue of: (**A**) Group II patients; (**B**) Group IIa; (**C**) Group IIb; (**D**) Group IIc.

**Figure 3 antibiotics-11-01747-f003:**
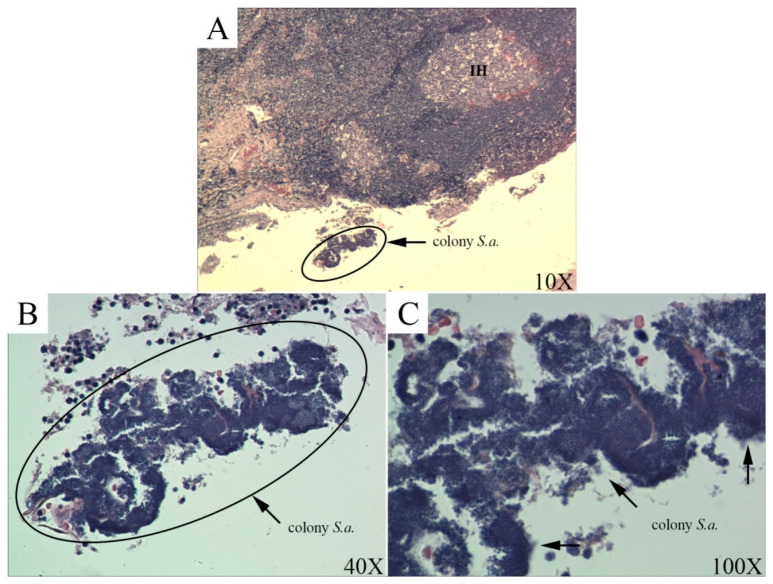
Histological examination (hematoxylin—eosin staining) of palatine tonsil tissue: (**A**) interfollicular hyperplasia, as a consequence of the inflammatory process; colony of *S. aureus*; (**B**,**C**) colony of *S. aureus*.

**Table 1 antibiotics-11-01747-t001:** Demographic data of 75 studied patients with chronic tonsillitis.

Demographic Data	Group I (*n*, %)	Group II (*n*, %)		
		Group IIa (*n*, %)	Group IIb (*n*, %)	Group IIc (*n*, %)
Gender				
M	12 (40)	11 (40.74)	4 (100)	8 (57.14)
F	18 (60)	16 (59.26)	-	6 (42.86)
Total number of participants	30 (100)	45 (100)		
		27 (100)	4 (100)	14 (100)
Age				
<5	-	6 (22.22)	2 (50)	8 (57.14)
5–10	-	19 (70.37)	2 (50)	6 (42.86)
11–16	-	2 (7.41)	-	-
17–20	4 (13.33)	-	-	-
21–30	16 (53.33)	-	-	-
31–40	10 (33.33)	-	-	-
Age (mean ± SD)	28.3 ± 6.7	5.56 ± 2.26		
		6.12 ± 2.4	5 ± 1.15	4.6 ± 1.65

**Table 2 antibiotics-11-01747-t002:** Clinical data of 75 studied patients with chronic tonsillitis.

CT Symptoms	Group I (n, %)	Group II (n, %)		
		Group IIa (*n*, %)	Group IIb (*n*, %)	Group IIc (*n*, %)
**Sore throat**				
No symptoms	-	24 (88.88)	4 (100)	
Very poorly expressed	-	1 (3.70)	-	
Poorly expressed	-	1 (3.70)	-	
Moderately severe	17 (56.66)	1 (3.70)	-	
Severe	13 (43.33)	-	-	
Strongly severe	-	-	-	
**Difficulty swallowing**				
No symptoms	-	25 (92.59)	4 (100)	
Very poorly expressed	-	-	-	
Poorly expressed	1 (3.33)	-	-	
Moderately severe	12 (40)	-	-	
Severe	17 (56.66)	2 (8.33)	-	
Strongly severe	-	-	-	
**Hard breathing**				
No symptoms		6 (22.22)	1 (25)	-
Very poorly expressed		2 (7.41)	-	-
Poorly expressed		1 (3.70)	1 (25)	1 (7.14)
Moderately severe		3 (11.11)	-	6 (42.86)
Severe		-	1 (25)	2 (14.29)
Strongly severe		15 (55.55)	1 (25)	5 (35.71)
**Secretion in the throat**				
No symptoms	4 (13.33)	11 (40.74)	-	
Very poorly expressed	-	1 (3.70)	1 (25)	
Poorly expressed	5 (16.66)	3 (11.11)	1 (25)	
Moderately severe	15 (50)	-	1 (25)	
Severe	6 (20)	1 (3.70)	1 (25)	
Strongly severe	-	11 (40.74)	-	
**Secretion in the nose**				
No symptoms	4 (13.33)	-	1 (25)	7 (50)
Very poorly expressed	-	-	-	-
Poorly expressed	4 (13.33)	-	1 (25)	2 (14.29)
Moderately severe	9 (30)	-	1 (25)	3 (21.43)
Severe	13 (43.33)	-	1 (25)	1 (7.14)
Strongly severe	-	-	-	1 (7.14)
**Cough**				
No symptoms	6 (20)	23 (85.18)	4 (100)	13 (92.86)
Very poorly expressed	2 (6.66)	2 (7.41)	-	-
Poorly expressed	6 (20)	-	-	-
Moderately severe	9 (30)	1 (3.70)	-	-
Severe	7 (23.33)	-	-	1 (7.14)
Strongly severe	-	1 (3.70)	-	-
**Bad breath**				
No symptoms	-	20 (70.07)	4 (100)	13 (92.86)
Very poorly expressed	-	3 (11.11)	-	-
Poorly expressed	-	-	-	1 (7.14)
Moderately severe	1 (3.33)	1 (3.70)	-	-
Severe	20 (66.66)	1 (3.70)	-	-
Strongly severe	9 (30)	2 (7.41)	-	-
**Snoring**				
No symptoms		11 (40.74)	1 (25)	4 (28.57)
Very poorly expressed		4 (14.81)	-	-
Poorly expressed		-	-	-
Moderately severe		2 (7.41)	-	-
Severe		1 (3.70)	-	6 (42.86)
Strongly severe		9 (33.33)	3 (75)	4 (28.57)
**Number of sore throat and ear inflammation**	4.33 ± 0.61	3.29 ± 1.92	4.5 ± 1	3.64 ± 2.31
**Antibiotic use**	30 (100)	17 (62.96)	3 (75)	8 (57.14)
**Oropharyngoscopic/Otoscopic * status**				
hyperemic, hypertrophic, cryptic tonsils with detritus in the crypts	13 (23.1)	25 (92.59)	4 (100)	-
asymmetric, uneven tonsils with hyperemia of the palatine arches	17 (56.67)	2 (7.41)	4 (100)	-
* adenoid hipertrophy	-	-	4 (100)	14 (100)
* blurred and non-transparent eardrum	-	-	4 (100)	11 (78.57)

* criteria for adenoidectomy.

**Table 3 antibiotics-11-01747-t003:** Bacteria isolated from palatine and adenoid tissue from patients with chronic tonsillitis.

Bacterial Species	Number of Isolates		
	Group I	Group II	
	Isolates from Palatine Tissue	Isolates from Palatine Tissue	Isolates from Adenoid Tissue
*Staphylococcus aureus*	10	5	2
*Staphylococcus warneri*	1	0	0
*Staphylococcus hominis*	1	0	0
*Streptococcus agalactiae*	1	0	1
*Streptococcus anginosus*	1	1	0
*Streptococcus constellatus*	0	2	0
*Streptococcus dysgalactiae*	2	0	0
*Streptococcus oralis*	7	15	5
*Streptococcus parasanquinis*	6	6	4
*Streptococcus pseudopneumoniae*	0	1	0
*Streptococcus pyogenes*	0	2	1
*Streptococcus salivarius*	0	1	0
*Micrococcus luteus*	0	1	0
*Rothia muciloginosa*	3	0	0
*Enterobacter cloacae*	2	0	0
*Stenotrophomonas maltophilia*	0	0	1
**Total number of isolated bacteria**	**35**	**33**	**14**

**Table 4 antibiotics-11-01747-t004:** Biofilm formation of *Staphylococcus aureus* isolates.

Biofilm—Producing Capacity	No. (%)
Weak	-
Moderate	2 (11.8%)
Strong	15 (88.2%)

**Table 5 antibiotics-11-01747-t005:** Antibiotic susceptibility among *Staphylococcus aureus* isolates from palatine and adenoid tissue (disk diffusion method, mm).

Antibiotics	*Staphylococcus aureus* Clinical Isolates (*n* = 17)	
	Resistance No (%)	Susceptibility No (%)
Penicillin G	17 (100)	0 (0)
Ampicillin	17 (100)	0 (0)
Amoxicillin	17 (100)	0 (0)
Amoxicillin with clavulanic acid	0 (0)	17 (100)
Cephalexin	0 (0)	17 (100)
Cefaclor	0 (0)	17 (100)
Cefpodoxime	0 (0)	17 (100)
Ceftriaxone	0 (0)	17 (100)
Ciprofloxacin	0 (0)	17 (100)
Erythromycin	2 (11.8)	15 (88.2)
Azithromycin	2 (11.8)	15 (88.2)
Roxithromycin	2 (11.8)	15 (88.2)
Clarithromycin	2 (11.8)	15 (88.2)
Clindamycin	2 (11.8)	15 (88.2)
Gentamicin	0 (0)	17 (100)
Chloramphenicol	0 (0)	17 (100)
Tetracycline	2 (11.8)	15 (88.2)

**Table 6 antibiotics-11-01747-t006:** Antibiotic susceptibility among *Staphylococcus aureus* isolates from palatine and adenoid tissue (microdilution method, mg/mL).

*Staphylococcus aureus* Isolates		Amoxicillin with Clavulanic Acid	Cefixime
1	MIC	0.003	0.003
	MBC	0.006	0.007
2	MIC	0.013	0.014
	MBC	0.026	0.028
3	MIC	0.013	0.014
	MBC	0.026	0.028
4	MIC	0.006	0.014
	MBC	0.013	0.007
5	MIC	0.001	0.003
	MBC	0.002	0.006
6	MIC	0.003	0.004
	MBC	0.006	0.007
7	MIC	0.001	0.002
	MBC	0.003	0.004
8	MIC	0.003	0.004
	MBC	0.006	0.007
9	MIC	0.001	0.002
	MBC	0.002	0.004
10	MIC	0.006	0.007
	MBC	0.006	0.014
11	MIC	0.003	0.003
	MBC	0.006	0.006
12	MIC	0.009	0.007
	MBC	0.013	0.014
13	MIC	0.005	0.005
	MBC	0.006	0.007
14	MIC	0.006	0.007
	MBC	0.010	0.008
15	MIC	0.013	0.014
	MBC	0.026	0.028
16	MIC	0.013	0.014
	MBC	0.026	0.028
17	MIC	0.003	0.003
	MBC	0.006	0.006

## Data Availability

Not applicable.

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
