# Peer review of "Analysis of Tonsil Tissues from Patients Diagnosed with Chronic Tonsillitis—Microbiological Profile, Biofilm-Forming Capacity and Histology"

_antibiotics, 2022, doi:10.3390/antibiotics11121747_

Round 1

Reviewer 1 Report

Dear authors,

The paper is interesting, but It's better that you complete you work analyzing the histomorphometric and clinical features of the mucosal biofilm in tonsil tissue of your patient, as performed by other authors.

Author Response

Response to Reviewer 1 Comments

Ponit 1: The paper is interesting, but It's better that you complete you work analyzing the histomorphometric and clinical features of the mucosal biofilm in tonsil tissue of your patient, as performed by other authors.

Response 1: Dear reviewer thank you for the kind comment. The aim of this work was only to histologically confirm the presence of bacteria in the tonsillar tissue. On our histological sections we didn’t detect the presence of the mucosal biofilm, so we can’t do histomorphometric and clinical significance of volume and thickness variations of mucosal biofilm in tonsil tissue as the Bulut et al., did in their article An analysis of the histomorphometric and clinical significance of mucosal biofilm in tonsil tissue of the children with a history of recurrent/chronic tonsillitis in both the mother and fatherˮ. But, we very appreciate your suggestions; it is a great idea for some different future research work that detects microbial biofilms in tonsillar tissue. Accordingly, we added a sentence in the results section, stating that microbial biofilm was not detected in the tonsillar tissue.

Reviewer 2 Report

The present work “Analysis of tonsil tissues from patients diagnosed with chronic 2 tonsillitis - microbiological profile, biofilm-forming capacity and histology” by Marina Kostić et al, appears a significant finding about the Chronic Tonsillitis (CT) conditions among various age groups of the population in Belgrade, Serbia. The present work identified the bacterial population in CT-infected palatine and adenoid tissues, and further categorized the bacteria on the basis of antibiotic susceptibility and biofilm-forming capacity. The present work nicely describes the CT in detail however, the work appears to be very restricted on the antibiotics susceptibility profiling of the bacteria (except disk diffusion test). The present work needs some significant inclusions to make it more insightful. Please check the comments and manuscript for details.  

Major comments-

· Table 3- Include the MALDI-ToF data to support the bacterium identification results

·   Table 5 Minimum Inhibitory Concentration (MIC) data is desired here to assess the resistance profile of the CT clinical isolates of S. aureus

·       Discuss the prevalence of S. aureus (a strong biofilm-forming bacteria) in adults versus S. oralis (no biofilm-forming bacteria) in children’s patients with CT. Do they affect the severity level? Please discuss in the discussion section with appropriate references.  

Minor comments-

·      Line 38-39- include some more relevant citations

·      Line 84-86- Introduce the role of other pathogenic bacteria than S. aureus in CT, please include appropriate citations.

·   Table 4- Please include the criteria used to differentiate the biofilm-forming ability of the bacteria below the table instead of putting it in the method section.

·   Line 215- Typo error- ‘paranasal cavities’ repeated

·   Line 237-238- discussion section- the role of enzymes such as beta-lactamases which drives the antibiotic resistance among the bacteria can be discussed in brief as well

Include biofilm images as a supplementary file

Author Response

Response to Reviewer 2 Comments

Dear reviewer thank you for the kind comments.

Major comments-

Point: 1 Table 3- Include the MALDI-ToF data to support the bacterium identification results

Response 1: We have added the MALDI-ToF data (confidentiality value) as a Supplementary file 1.

Point 2: Table 5 Minimum Inhibitory Concentration (MIC) data is desired here to assess the resistance profile of the CT clinical isolates of S. aureus

Response 2: We have added the following information as a Table 6. We have added the following paragraph:

As we mentioned in Section 2.2, all patients took medications from the group of beta lactam antibiotics and cephalosporin’s as part of drug therapy. Based on questionnaire the most prescribed drugs for the CT treatment were amoxicillin with clavulanic acid (beta lactam antibiotic) and cefixime (cephalosporin). Therefore, these two antibiotics were chosen for further susceptibility testing (Table 6). S. aureus isolates proved to be more sensitive to amoxicillin with clavulanic acid with MIC values in range 0.001 – 0.013 mg/mL (Table 6.). The MIC/MBC values for cefixime were slightly higher (0.003-0.014 mg/mL / 0.006-0.028 mg/mL, respectively).

Also, we have added the following paragraph in Materials and Methods, section 4.5:

… and microdillution method [36]. For microdilution method, the results were presented as minimum inhibitory/bactericidal concentrations (MICs/MBCs). Amoxicillin with clavulanic acid (Hemofarm, Vršac, Serbia) and cefixime (Alkaloid, Skoplje, Macedonia) were used as positive controls.

Point 3: Discuss the prevalence of S. aureus (a strong biofilm-forming bacteria) in adults versus S. oralis (no biofilm-forming bacteria) in children’s patients with CT. Do they affect the severity level? Please discuss in the discussion section with appropriate references.

Response 3: We have added the following paragraph:

Our results showed prevalence of S. oralis in the children which can be explained by the outbreaks of tonsillitis which is common in children of all ages, especially in the school-aged children in which is crowding common [32]. Also, S. oralis is known to be the first to colonize enamel, often interacting with periodontal pathogens such as Porphyromonas gingivalis which is the causative agent of chronic periodontitis [33]. Repeated and untreated Streptoccocal infection can lead to the colonization of some other pathogenic bacteria such as S. aureus in adults (Table 3). Also, in immunocompromised patients, it can cause bacterial endocarditis, respiratory diseases, and can also cause streptococcal septicemia [34].

Minor comments-

  •     Line 38-39- include some more relevant citations

The following citations are included:

Esposito, S.; Principi, N. Impact of Nasopharyngeal Microbiota on the Development of Respiratory Tract Diseases. Eur. J. Clin. Microbiol. Infect. Dis. 2017 371 2017, 37, 1–7, doi:10.1007/S10096-017-3076-7.

Johnson, M.M. Ear, Nose, and Throat Infections; Second Edi.; Elsevier Inc., 2018; ISBN 9780323445856.

Bosch, A.A.T.M.; Biesbroek, G.; Trzcinski, K.; Sanders, E.A.M.; Bogaert, D. Viral and Bacterial Interactions in the Upper Respiratory Tract. PLoS Pathog. 2013, 9, doi:10.1371/journal.ppat.1003057.

Claassen-Weitz, S.; Lim, K.Y.L.; Mullally, C.; Zar, H.J.; Nicol, M.P. Systematic Review The Association between Bacteria Colonizing the Upper Respiratory Tract and Lower Respiratory Tract Infection in Young Children: A Systematic Review and Meta-Analysis. Clin. Microbiol. Infect. 2021, 27, 1262–1270, doi:10.1016/j.cmi.2021.05.034.

  • Line 84-86- Introduce the role of other pathogenic bacteria than S. aureus in CT, please include appropriate citations.

We have added the following paragraph: The tonsillar tissue and adenoids are predisposed to biofilm formation due to cryptic tissue structure and direct, repeated exposure to respiratory bacterial pathogens [16]. Except S. aureus, several other respiratory pathogens such as Haemophilus influenzae, Klebsiella pneumoniae and Streptococcus pneumoniae, can persists predominantly intracellularly in the form of biofilm in adenoids and palatine tonsils [2,16,20].

  • Table 4- Please include the criteria used to differentiate the biofilm-forming ability of the bacteria below the table instead of putting it in the method section.

Interpretation of biofilm production: OD ≤ ODc – none; ODc ˂ OD ≤ 2 × ODc - weak; 2 × ODc ˂ OD ≤ 4 × ODc - moderate; 4 × ODc ˂ OD – strong.

We have included this explanation now below the table.

  •  Line 215- Typo error- ‘paranasal cavities’ repeated

This was corrected.

  •  Line 237-238- discussion section- the role of enzymes such as beta-lactamases which drives the antibiotic resistance among the bacteria can be discussed in brief as well

We have added the following paragraph:

Also, one of key factor to antibiotic resistance against penicillin group of antibiotics is beta-lactamase production. These enzymes are produced by Gram (+) bacteria extracellularly, and by Gram (-) bacteria in the periplasmic space. β-lactamases can inactivate almost all β-lactam antibiotics by binding covalently to their carbonyl moiety and hydrolyzing the β-lactam ring [30].

Include biofilm images as a supplementary file.

We have included the biofilm images as a Supplementary file 2.

Round 2

Reviewer 1 Report

Dear Authors,

thanks for your answer.

Reviewer 2 Report

The authors have done the revision very well in a short period of time. In my opinion, the revised manuscript makes more sense now. Hence, the revised manuscript can be published. 

Thanks